# High Dimensional Linear Regression
# using Lattice Basis Reduction

**David Gamarnik**
Sloan School of Management
Massachussetts Institute of Technology
Cambridge, MA 02139
gamarnik@mit.edu

**Ilias Zadik**
Operations Research Center
Massachussetts Institute of Technology
Cambridge, MA 02139
izadik@mit.edu

## Abstract

We consider a high dimensional linear regression problem where the goal is to efficiently recover an unknown vector $\beta^*$ from $n$ noisy linear observations $Y = X\beta^* + W \in \mathbb{R}^n$, for known $X \in \mathbb{R}^{n \times p}$ and unknown $W \in \mathbb{R}^n$. Unlike most of the literature on this model we make no sparsity assumption on $\beta^*$. Instead we adopt a regularization based on assuming that the underlying vectors $\beta^*$ have rational entries with the same denominator $Q \in \mathbb{Z}_{>0}$. We call this $Q$-rationality assumption. We propose a new polynomial-time algorithm for this task which is based on the seminal Lenstra-Lenstra-Lovasz (LLL) lattice basis reduction algorithm. We establish that under the $Q$-rationality assumption, our algorithm recovers exactly the vector $\beta^*$ for a large class of distributions for the iid entries of $X$ and non-zero noise $W$. We prove that it is successful under small noise, even when the learner has access to only one observation ($n = 1$). Furthermore, we prove that in the case of the Gaussian white noise for $W$, $n = o\left(p/\log p\right)$ and $Q$ sufficiently large, our algorithm tolerates a nearly optimal information-theoretic level of the noise.

## 1 Introduction

We consider the following high-dimensional linear regression model. Consider $n$ samples of a vector $\beta^* \in \mathbb{R}^p$ in a vector form $Y = X\beta^* + W$ for some $X \in \mathbb{R}^{n \times p}$ and $W \in \mathbb{R}^n$. Given the knowledge of $Y$ and $X$ the goal is to infer $\beta^*$ using an efficient algorithm and the minimum number $n$ of samples possible. Throughout the paper we call $p$ the number of features, $X$ the measurement matrix and $W$ the noise vector.

We focus on the high-dimensional case where $n$ may be much smaller than $p$ and $p$ grows to infinity, a setting that has been very popular in the literature during the last years Chen et al. (2001), Donoho (2006), Candes et al. (2006), Foucart and Rauhut (2013), Wainwright (2009). In this case, and under no additional structural assumption, the inference task becomes impossible, even in the noiseless case $W = 0$, as the underlying linear system becomes underdetermined. Most papers address this issue by imposing a *sparsity assumption* on $\beta^*$, which refers to $\beta^*$ having only a limited number of non-zero entries compared to its dimension Donoho (2006), Candes et al. (2006), Foucart and Rauhut (2013). During the past decades, the sparsity assumption led to a fascinating line of research in statistics and compressed sensing, which established, among other results, that several polynomial-time algorithms, such as Basis Pursuit Denoising Scheme and LASSO, can efficiently recover a sparse $\beta^*$ with number of samples much smaller than the number of features Candes et al. (2006), Wainwright (2009), Foucart and Rauhut (2013). For example, it is established that if $\beta^*$ is constrained to have at most $k \leq p$ non-zero entries, $X$ has iid $N(0, 1)$ entries, $W$ has iid $N(0, \sigma^2)$ entries and $n$ is of the order $k \log\left(\frac{p}{k}\right)$, then both of the mentioned algorithms can recover $\beta^*$, up to the level of the noise. Different

structural assumptions than sparsity have also been considered in the literature. For example, a recent paper Bora et al. (2017) makes the assumption that $\beta^*$ lies near the range of an $L$-Lipschitz generative model $G : \mathbb{R}^k \to \mathbb{R}^p$ and it proposes an algorithm which succeeds with $n = O(k \log L)$ samples.

A downside of all of the above results is that they provide no guarantee in the case $n$ is much smaller than $k \log \left(\frac{p}{k}\right)$. Consider for example the case where the components of a sparse $\beta^*$ are binary-valued, and $X, W$ follow the Gaussian assumptions described above. Supposing that $\sigma$ is sufficiently small, it is a straightforward argument that even when $n = 1$, $\beta^*$ is recoverable from $Y = \langle X, \beta^* \rangle + W$ by a brute-force method with probability tending to one as $p$ goes to infinity (whp). On the other hand, for sparse and binary-valued $\beta^*$, the Basis Pursuit method in the noiseless case Donoho and Tanner (2006) and the Basis Pursuit Denoising Scheme in the noisy case Gamarnik and Zadik (2017b) have been proven to fail to recover a binary $\beta^*$ with $n = o(k \log \left(\frac{p}{k}\right))$ samples. Furthermore, LASSO has been proven to fail to recover a vector with the same support of $\beta^*$, with $n = o(k \log p)$ samples Wainwright (2009). This failure to capture the complexity of the problem accurately enough for small sample sizes also lead to an algorithmic hardness conjecture for the regime $n = o(k \log \left(\frac{p}{k}\right))$ Gamarnik and Zadik (2017a), Gamarnik and Zadik (2017b). While this conjecture still stands in the general case, as we show in this paper, in the special case where $\beta^*$ is rational-valued and the magnitude of the noise $W$ is sufficiently small, the statistical computational gap can be closed and $\beta^*$ can be recovered even when $n = 1$.

The structural assumption we impose on $\beta^*$ is that its entries are rational numbers with denominator equal to some fixed positive integer value $Q \in \mathbb{Z}_{>0}$, something we refer to as the *Q-rationality assumption*. Note that for any $Q$, this assumption is trivially satisfied by the binary-valued $\beta^*$ which was discussed above. The 1-rationality assumption corresponds to $\beta^*$ having integer entries, which is well-motivated in practise. For example, this assumption appears frequently in the study of global navigation satellite systems (GPS) and communications Hassibi and Boyd (1998), Hassibi and Vikalo (2002), Brunel and Boutros (1999), Borno (2011). In the first reference the authors propose a mixed linear/integer model of the form $Y = Ax + Bz + W$ where z is an integer valued vector corresponding to integer multiples of certain wavelength. Several examples corresponding to regression models with integer valued regression coefficients and zero noise (though not always in the same model) are also discussed in the book Foucart and Rauhut (2013). In particular one application is the so-called Single-Pixel camera. In this model a vector $\beta$ corresponds to color intensities of an image for different pixels and thus takes discrete values. The model assumes no noise, which is one of the assumptions we adopt in our model, though the corresponding regression matrix has i.i.d. $+1/-1$ Bernoulli entries, as opposed to a continuous distribution we assume. Two other applications involving noiseless regression models found in the same reference are MRI imaging and Radar detection.

A large body of literature on noiseless regression type models is a series of papers on phase retrieval. Here the coefficients of the regression vector $\beta^*$ and the entries of the regression matrix $X$ are complex valued, but the observation vector $Y = X\beta^*$ is only observed through absolute values. This model has many applications, including crystallography, see Candes et al. (2015). The aforementioned paper provides many references to phase retrieval model including the cases when the entries of $\beta^*$ have a finite support. We believe that our method can also be extended so that to model the case where the entries of the regression vector have a finite support, even if irrationally valued, and the entries of $Y$ are only observed through their magnitude. In other words, we expect that the method of the present paper applies to the phase retrieval problem at least in some of the cases and this is one of the current directions we are exploring.

Noiseless regression model with integer valued regression coefficients were also important in the theoretical development of compressive sensing methods. Specifically, Donoho Donoho (2006) and Donoho and Tanner Donoho and Tanner (2005),Donoho and Tanner (2006),Donoho and Tanner (2009) consider a noiseless regression model of the form $AB$ where $A$ is a random (say Gaussian) matrix and $B$ is the unit cube $[0, 1]^p$. One of the goals of these papers was to count number of extreme points of the projected polytope $AB$ in order to explain the effectiveness of the linear programming based methods. The extreme points of this polytope can only appear as projections of extreme points of $B$ which are all length-p binary vector, namely one deals with noiseless regression model with binary coefficients – an important special case of the model we consider in our paper.

In the Bayesian setting, where the ground truth $\beta^*$ is sampled according to a discrete distribution Donoho et al. (2013) proposes a low-complexity algorithm which provably recovers $\beta^*$ with $n = o(p)$

samples. This algorithm uses the technique of approximate message passing (AMP) and is motivated by ideas from statistical physics Krzakala et al. (2012). Even though the result from Donoho et al. (2013) applies to the general discrete case for $\beta^*$, it requires the matrix $X$ to be spatially coupled, a property that in particular does not hold for $X$ with iid standard Gaussian entries. Furthermore the required sample size for the algorithm to work is only guaranteed to be sublinear in $p$, a sample size potentially much bigger than the information-theoretic limit for recovery under sufficiently small noise ($n = 1$). In the present paper, where $\beta^*$ satisfies the $Q$-rationality assumption, we propose a polynomial-time algorithm which applies for a large class of continuous distributions for the iid entries of $X$, including the normal distribution, and provably works even when $n = 1$.

The algorithm we propose is inspired by the algorithm introduced in Lagarias and Odlyzko (1985) which solves, in polynomial time, a certain version of the so-called Subset-Sum problem. To be more specific, consider the following NP-hard algorithmic problem. Given $p \in \mathbb{Z}_{>0}$ and $y, x_1, x_2, \ldots, x_p \in \mathbb{Z}_{>0}$ the goal is to find a $\emptyset \neq S \subset [p]$ with $y = \sum_{i \in S} x_i$ when at least one such set $S$ is assumed to exist. Over 30 years ago, this problem received a lot of attention in the field of cryptography, based on the belief that the problem would be hard to solve in many "real" instances. This would imply that several already built public key cryptosystems, called knapsack public key cryptosystems, could be considered safe from attacks Lempel (1979), Merkle and Hellman (1978). This belief though was proven wrong by several papers in the early 80s, see for example Shamir (1982). Motivated by this line of research, Lagarias and Odlyzko in Lagarias and Odlyzko (1985), and a year later Frieze in Frieze (1986), using a cleaner and shorter argument, proved the same surprising fact: if $x_1, x_2, \ldots, x_p$ follow an iid uniform distribution on $[2^{\frac{1}{2}(1+\epsilon)p^2}] := \{1, 2, 3, \ldots, 2^{\frac{1}{2}(1+\epsilon)p^2}\}$ for some $\epsilon > 0$ then there exists a polynomial-in-$p$ time algorithm which solves the subset-sum problem whp as $p \to +\infty$. In other words, even though the problem is NP-hard in the worst-case, assuming a quadratic-in-$p$ number of bits for the coordinates of $x$, the algorithmic complexity of the typical such problem is polynomial in $p$. The successful efficient algorithm is based on an elegant application of a seminal algorithm in the computational study of lattices called the Lenstra-Lenstra-Lovasz (LLL) algorithm, introduced in Lenstra et al. (1982). This algorithm receives as an input a basis $\{b_1, \ldots, b_m\} \subset \mathbb{Z}^m$ of a full-dimensional lattice $\mathcal{L}$ and returns in time polynomial in $m$ and $\max_{i=1,2,\ldots,m} \log \|b_i\|_\infty$ a non-zero vector $\hat{z}$ in the lattice, such that $\|\hat{z}\|_2 \leq 2^{\frac{m}{2}} \|z\|_2$, for all $z \in \mathcal{L} \setminus \{0\}$.

Besides its significance in cryptography, the result of Lagarias and Odlyzko (1985) and Frieze (1986) enjoys an interesting linear regression interpretation as well. One can show that under the iid uniform in $[2^{\frac{1}{2}(1+\epsilon)p^2}]$ assumption for $x_1, x_2, \ldots, x_p$, there exists exactly one set $S$ with $y = \sum_{i \in S} x_i$ whp as $p$ tends to infinity. Therefore if $\beta^*$ is the indicator vector of this unique set $S$, that is $\beta_i^* = 1(i \in S)$ for $i = 1, 2, \ldots, p$, we have that $y = \sum_i x_i \beta_i^* = \langle x, \beta^* \rangle$ where $x := (x_1, x_2, \ldots, x_p)$. Furthermore using only the knowledge of $y, x$ as input to the Lagarias-Odlyzko algorithm we obtain a polynomial in $p$ time algorithm which recovers exactly $\beta^*$ whp as $p \to +\infty$. Written in this form, and given our earlier discussion on high-dimensional linear regression, this statement is equivalent to the statement that the noiseless high-dimensional linear regression problem with binary $\beta^*$ and $X$ generated with iid elements from $\mathrm{Unif}[2^{\frac{1}{2}(1+\epsilon)p^2}]$ is polynomial-time solvable even with one sample ($n = 1$), whp as $p$ grows to infinity. The main focus of this paper is to extend this result to $\beta^*$ satisfying the $Q$-rationality assumption, continuous distributions on the iid entries of $X$ and non-trivial noise levels.

**Summary of the Results**

We propose a polynomial time algorithm for high-dimensional linear regression problem and establish a general result for its performance. We show that if the entries of $X \in \mathbb{R}^{n \times p}$ are iid from an arbitrary continuous distribution with bounded density and finite expected value, $\beta^*$ satisfies the $Q$-rationality assumption, $\|\beta^*\|_\infty \leq R$ for some $R > 0$, and $W$ is either an adversarial vector with infinity norm at most $\sigma$ or has iid mean-zero entries with variance at most $\sigma^2$, then under some explicitly stated assumption on the parameters $n, p, \sigma, R, Q$ our algorithm recovers exactly the vector $\beta^*$ in time which is polynomial in $n, p, \log(\frac{1}{\sigma}), \log R, \log Q$, whp as $p$ tends to infinity. As a corollary, we show that for any $Q$ and $R$ our algorithm can infer correctly $\beta^*$, when $\sigma$ is at most exponential in $-\left(p^2/2 + (2 + p) \log(QR)\right)$, even from one observation ($n = 1$). We show that for general $n$ our algorithm can tolerate noise level $\sigma$ which is exponential in $-\left((2n + p)^2/2n + (2 + p/n) \log(QR)\right)$. We complement our results with the information-theoretic limits of our problem. We show that in the case of Gaussian white noise $W$, a noise level which is exponential in $-\frac{p}{n} \log(QR)$, which is

essentially the second part of our upper bound, cannot be tolerated. This allows us to conclude that in the regime $n = o\left(p/\log p\right)$ and $RQ = 2^{\omega(p)}$ our algorithm tolerates the optimal information theoretic level of noise.

The algorithm we propose receives as input real-valued data $Y, X$ but importantly it truncates in the first step the data by keeping the first $N$ bits after zero of every entry. In particular, this allows the algorithm to perform only **finite-precision** arithmetic operations. Here $N$ is a parameter of our algorithm chosen by the algorithm designer. For our recovery results it is chosen to be polynomial in $p$ and $\log(\frac{1}{\sigma})$.

A crucial step towards our main result is the extension of the Lagarias-Odlyzko algorithm Lagarias and Odlyzko (1985), Frieze (1986) to not necessarily binary, integer vectors $\beta^* \in \mathbb{Z}^p$, for measurement matrix $X \in \mathbb{Z}^{n \times p}$ with iid entries not necessarily from the uniform distribution, and finally, for non-zero noise vector $W$. As in Lagarias and Odlyzko (1985) and Frieze (1986), the algorithm we construct depends crucially on building an appropriate lattice and applying the LLL algorithm on it. There is though an important additional step in the algorithm presented in the present paper compared with the algorithm in Lagarias and Odlyzko (1985) and Frieze (1986). The latter algorithm is proven to recover a non-zero integer multiple $\lambda\beta^*$ of the underlying binary vector $\beta^*$. Then since $\beta^*$ is known to be binary, the exact recovery becomes a matter of renormalizing out the factor $\lambda$ from every non-zero coordinate. On the other hand, even if we establish in our case the corresponding result and recover a non-zero integer multiple of $\beta^*$ whp, this last renormalizing step would be impossible as the ground truth vector is not assumed to be binary. We address this issue as follows. First we notice that the renormalization step remains valid if the greatest common divisor of the elements of $\beta^*$ is 1. Under this assumption from any non-zero integer multiple of $\beta^*$, $\lambda\beta^*$ we can obtain the vector itself by observing that the greatest common divisor of $\lambda\beta^*$ equals to $\lambda$, and computing $\lambda$ by using for instance the Euclid's algorithm. We then generalize our recovery guarantee to arbitrary $\beta^*$. We do this by first translating implicitly the vector $\beta^*$ with a random integer vector $Z$ via translating our observations $Y = X\beta^* + W$ by $XZ$ to obtain $Y + XZ = X(\beta^* + Z) + W$. We then prove that the elements of $\beta^* + Z$ have greatest common divisor equal to unity with probability tending to one. This last step is based on an analytic number theory argument which slightly extends a beautiful result from probabilistic number theory (see for example, Theorem 332 in Hardy and Wright (1975)) according to which $\lim_{m\to+\infty} \mathbb{P}_{P,Q\sim\text{Unif}\{1,2,\dots,m\},P\perp Q}\left[\gcd\left(P,Q\right) = 1\right] = \frac{6}{\pi^2}$, where $P \perp Q$ refers to $P, Q$ being independent random variables. This result is not of clear origin in the literature, but possibly it is attributed to Chebyshev, as mentioned in Erdos and Lorentz (1985). A key implication of this result for us is the fact that the limit above is strictly positive.

**Notation**

Let $\mathbb{Z}^*$ denote $\mathbb{Z} \setminus \{0\}$. For $k \in \mathbb{Z}_{>0}$ we set $[k] := \{1, 2, \dots, k\}$. For a vector $x \in \mathbb{R}^d$ we define $\text{Diag}_{d\times d}(x) \in \mathbb{R}^{d\times d}$ to be the diagonal matrix with $\text{Diag}_{d\times d}(x)_{ii} = x_i$, for $i \in [d]$. For $1 \le p < \infty$ by $\mathcal{L}_p$ we refer to the standard $p$-norm notation for finite dimensionall real vectors. Given two vectors $x, y \in \mathbb{R}^d$ the Euclidean inner product notation is denoted by $\langle x, y \rangle := \sum_{i=1}^d x_i y_i$. By $\log : \mathbb{R}_{>0} \to \mathbb{R}$ we refer the logarithm with base 2. The lattice $\mathcal{L} \subseteq \mathbb{Z}^k$ generated by a set of linearly independent $b_1, \dots, b_k \in \mathbb{Z}^k$ is defined as $\{\sum_{i=1}^k z_i b_i | z_1, z_2, \dots, z_k \in \mathbb{Z}\}$. Throughout the paper we use the standard asymptotic notation, $o, O, \Theta, \Omega$ for comparing the growth of two real-valued sequences $a_n, b_n, n \in \mathbb{Z}_{>0}$. Finally, we say that a sequence of events $\{A_p\}_{p\in\mathbb{N}}$ holds with high probability (whp) as $p \to +\infty$ if $\lim_{p\to+\infty} \mathbb{P}\left(A_p\right) = 1$.

## 2 Main Results

### 2.1 Extended Lagarias-Odlyzko algorithm

Let $n, p, R \in \mathbb{Z}_{>0}$. Given $X \in \mathbb{Z}^{n\times p}, \beta^* \in \left(\mathbb{Z} \cap [-R, R]\right)^p$ and $W \in \mathbb{Z}^n$, set $Y = X\beta^* + W$. From the knowledge of $Y, X$ the goal is to infer exactly $\beta^*$. For this task we propose the following algorithm which is an extension of the algorithm in Lagarias and Odlyzko (1985) and Frieze (1986). For realistic purposes the values of $R, \|W\|_\infty$ is not assumed to be known exactly. As a result, the following algorithm, besides $Y, X$, receives as an input a number $\hat{R} \in \mathbb{Z}_{>0}$ which is an estimated

upper bound in absolute value for the entries of $\beta^*$ and a number $\hat{W} \in \mathbb{Z}_{>0}$ which is an estimated upper bound in absolute value for the entries of $W$.

---

**Algorithm 1** Extended Lagarias-Odlyzko (ELO) Algorithm

---

**Input:** $(Y, X, \hat{R}, \hat{W})$, $Y \in \mathbb{Z}^n$, $X \in \mathbb{Z}^{n \times p}$, $\hat{R}, \hat{W} \in \mathbb{Z}_{>0}$.

**Output:** $\hat{\beta}^*$ an estimate of $\beta^*$

1   Generate a random vector $Z \in \{\hat{R} + 1, \hat{R} + 2, \ldots, 2\hat{R} + \log p\}^p$ with iid entries uniform in $\{\hat{R} + 1, \hat{R} + 2, \ldots, 2\hat{R} + \log p\}$

2   Set $Y_1 = Y + XZ$.

3   For each $i = 1, 2, \ldots, n$, if $|(Y_1)_i| < 3$ set $(Y_2)_i = 3$ and otherwise set $(Y_2)_i = (Y_1)_i$.

4   Set $m = 2^{n + \lceil \frac{p}{2} \rceil + 3} p \left( \hat{R} \lceil \sqrt{p} \rceil + \hat{W} \lceil \sqrt{n} \rceil \right)$.

5   Output $\hat{z} \in \mathbb{R}^{2n+p}$ from running the LLL basis reduction algorithm on the lattice generated by the columns of the following $(2n + p) \times (2n + p)$ integer-valued matrix,

$$
A_m := \left[ \begin{array}{ccc} mX & -m\mathrm{Diag}_{n \times n}(Y_2) & mI_{n \times n} \\ I_{p \times p} & 0_{p \times n} & 0_{p \times n} \\ 0_{n \times p} & 0_{n \times n} & I_{n \times n} \end{array} \right] \tag{1}
$$

6   Compute $g = \gcd(\hat{z}_{n+1}, \hat{z}_{n+2}, \ldots, \hat{z}_{n+p})$, using the Euclid's algorithm.

7   If $g \neq 0$, output $\hat{\beta}^* = \frac{1}{g}(\hat{z}_{n+1}, \hat{z}_{n+2}, \ldots, \hat{z}_{n+p})^t - Z$. Otherwise, output $\hat{\beta}^* = 0_{p \times 1}$.

---

We explain here informally the steps of the (ELO) algorithm and briefly sketch the motivation behind each one of them. In the first and second steps the algorithm translates $Y$ by $XZ$ where $Z$ is a random vector with iid elements chosen uniformly from $\{\hat{R} + 1, \hat{R} + 2, \ldots, 2\hat{R} + \log p\}$. In that way $\beta^*$ is translated implicitly to $\beta = \beta^* + Z$ because $Y_1 = Y + XZ = X(\beta^* + Z) + W$. As we will establish using a number theoretic argument, $\gcd(\beta) = 1$ whp as $p \to +\infty$ with respect to the randomness of $Z$, even though this is not necessarily the case for the original $\beta^*$. This is an essential requirement for our technique to exactly recover $\beta^*$ and steps six and seven to be meaningful. In the third step the algorithm gets rid of the significantly small observations. The minor but necessary modification of the noise level affects the observations in a negligible way.

The fourth and fifth steps of the algorithm provide a basis for a specific lattice in $2n + p$ dimensions. The lattice is built with the knowledge of the input and $Y_2$, the modified $Y$. The algorithm in step five calls the LLL basis reduction algorithm to run for the columns of $A_m$ as initial basis for the lattice. The fact that $Y$ has been modified to be non-zero on every coordinate is essential here so that $A_m$ is full-rank and the LLL basis reduction algorithm, defined in Lenstra et al. (1982), can be applied,. This application of the LLL basis reduction algorithm is similar to the one used in Frieze (1986) with one important modification. In order to deal here with multiple equations and non-zero noise, we use $2n + p$ dimensions instead of $1 + p$ in Frieze (1986). Following though a similar strategy as in Frieze (1986), it can be established that the $n + 1$ to $n + p$ coordinates of the output of the algorithm, $\hat{z} \in \mathbb{Z}^{2n+p}$, correspond to a vector which is a non-zero integer multiple of $\beta$, say $\lambda\beta$ for $\lambda \in \mathbb{Z}^*$, w.h.p. as $p \to +\infty$.

The proof of the above result is an important part in the analysis of the algorithm and it is heavily based on the fact that the matrix $A_m$, which generates the lattice, has its first $n$ rows multiplied by the "large enough" and appropriately chosen integer $m$ which is defined in step four. It can be shown that this property of $A_m$ implies that any vector $z$ in the lattice with "small enough" $\mathcal{L}_2$ norm necessarily satisfies $(z_{n+1}, z_{n+2}, \ldots, z_{n+p}) = \lambda\beta$ for some $\lambda \in \mathbb{Z}^*$ whp as $p \to +\infty$. In particular, using that $\hat{z}$ is guaranteed to satisfy $\|\hat{z}\|_2 \leq 2^{\frac{2n+p}{2}} \|z\|_2$ for all non-zero $z$ in the lattice, it can be derived that $\hat{z}$ has a "small enough" $\mathcal{L}_2$ norm and therefore indeed satisfies the desired property whp as $p \to +\infty$. Assuming now the validity of the $\gcd(\beta) = 1$ property, step six finds in polynomial time this unknown integer $\lambda$ that corresponds to $\hat{z}$, because $\gcd(\hat{z}_{n+1}, \hat{z}_{n+2}, \ldots, \hat{z}_{n+p}) = \gcd(\lambda\beta) = \lambda$. Finally step seven scales out $\lambda$ from every coordinate and then subtracts the known random vector $Z$, to output exactly $\beta^*$.

Of course the above is based on an informal reasoning. Formally we establish the following result.

**Theorem 2.1.** *Suppose*

(1) $X \in \mathbb{Z}^{n \times p}$ is a matrix with iid entries generated according to a distribution $\mathcal{D}$ on $\mathbb{Z}$ which for some $N \in \mathbb{Z}_{>0}$ and constants $C, c > 0$, assigns at most $\frac{c}{2^N}$ probability on each element of $\mathbb{Z}$ and satisfies $\mathbb{E}[|V|] \leq C2^N$, for $V \stackrel{d}{=} \mathcal{D}$;

(2) $\beta^* \in (\mathbb{Z} \cap [-R, R])^p$, $W \in \mathbb{Z}^n$;

(3) $Y = X\beta^* + W$.

Suppose furthermore that $\hat{R} \geq R$ and

$$N \geq \frac{1}{2n}(2n + p) \left[ 2n + p + 10 \log \left( \hat{R}\sqrt{p} + (\|W\|_\infty + 1)\sqrt{n} \right) \right] + 6 \log \left( (1 + c)\, np \right). \quad (2)$$

For any $\hat{W} \geq \|W\|_\infty$ the algorithm ELO with input $(Y, X, \hat{R}, \hat{W})$ outputs **exactly** $\beta^*$ w.p. $1 - O\left(\frac{1}{np}\right)$ (whp as $p \to +\infty$) and terminates in time at most polynomial in $n, p, N, \log \hat{R}$ and $\log \hat{W}$.

**Remark 2.2.** *In the statement of Theorem 2.1 the only parameters that are assumed to grow to infinity are $p$ and whichever other parameters among $n, R, \|W\|_\infty, N$ are implied to grow to infinity because of (2). Note in particular that $n$ can remain bounded, including the case $n = 1$, if $N$ grows fast enough.*

**Remark 2.3.** *It can be easily checked that the assumptions of Theorem 2.1 are satisfied for $n = 1$, $N = (1 + \epsilon)\frac{p^2}{2}$, $R = 1$, $\mathcal{D} = \text{Unif}\{1, 2, 3, \ldots, 2^{(1+\epsilon)\frac{p^2}{2}}\}$ and $W = 0$. Under these assumptions, the Theorem's implication is a generalization of the result from Lagarias and Odlyzko (1985) and Frieze (1986) to the case $\beta^* \in \{-1, 0, 1\}^p$.*

## 2.2 Applications to High-Dimensional Linear Regression

### The Model

We first define the $Q$-rationality assumption.

**Definition 2.4.** *Let $p, Q \in \mathbb{Z}_{>0}$. We say that a vector $\beta \in \mathbb{R}^p$ satisfies the Q-**rationality assumption** if for all $i \in [p]$, $\beta_i^* = \frac{K_i}{Q}$, for some $K_i \in \mathbb{Z}$.*

The high-dimensional linear regression model we are considering is as follows.

**Assumptions 1.** *Let $n, p, Q \in \mathbb{Z}_{>0}$ and $R, \sigma, c > 0$. Suppose*

(1) *measurement matrix $X \in \mathbb{R}^{n \times p}$ with iid entries generated according to a continuous distribution $\mathcal{C}$ which has density $f$ with $\|f\|_\infty \leq c$ and satisfies $\mathbb{E}[|V|] < +\infty$, where $V \stackrel{d}{=} \mathcal{C}$;*

(2) *ground truth vector $\beta^*$ satisfies $\beta^* \in [-R, R]^p$ and the Q-rationality assumption;*

(3) *$Y = X\beta^* + W$ for some noise vector $W \in \mathbb{R}^n$. It is assumed that either $\|W\|_\infty \leq \sigma$ or $W$ has iid entries with mean zero and variance at most $\sigma^2$, depending on the context.*

**Objective:** Based on the knowledge of $Y$ and $X$ the goal is to recover $\beta^*$ using an efficient algorithm and using the smallest number $n$ of samples possible. The recovery should occur with high probability (w.h.p), as $p$ diverges to infinity.

### The Lattice-Based Regression (LBR) Algorithm

As mentioned in the Introduction, we propose an algorithm to solve the regression problem, which we call the Lattice-Based Regression (LBR) algorithm. The exact knowledge of $Q, R, \|W\|_\infty$ is not assumed. Instead the algorithm receives as an input, additional to $Y$ and $X$, $\hat{Q} \in \mathbb{Z}_{>0}$ which is an estimated multiple of $Q$, $\hat{R} \in \mathbb{Z}_{>0}$ which is an estimated upper bound in absolute value for the entries of $\beta^*$ and $\hat{W} \in \mathbb{R}_{>0}$ which is an estimated upper bound in absolute value for the entries of the noise vector $W$. Furthermore an integer number $N \in \mathbb{Z}_{>0}$ is given to the algorithm as an input, which, as we will explain, corresponds to a truncation in the data in the first step of the algorithm.

---
**Algorithm 2** Lattice Based Regression (LBR) Algorithm
---
**Input:** $(Y, X, N, \hat{Q}, \hat{R}, \hat{W})$, $Y \in \mathbb{Z}^n$, $X \in \mathbb{Z}^{n \times p}$ and $N, \hat{Q}, \hat{R}, \hat{W} \in \mathbb{Z}_{>0}$.

**Output:** $\hat{\beta}^*$ an estimate of $\beta^*$

**8** Set $Y_N = ((Y_i)_N)_{i \in [n]}$ and $X_N = ((X_{ij})_N)_{i \in [n], j \in [p]}$.

**9** Set $(\hat{\beta}_1)^*$ to be the output of the ELO algorithm with input:

$$\left( 2^N \hat{Q} Y_N, 2^N X_N, \hat{Q}\hat{R}, 2\hat{Q}\left( 2^N \hat{W} + \hat{R}p \right) \right).$$

**10** Output $\hat{\beta}^* = \frac{1}{\hat{Q}}(\hat{\beta}_1)^*$.

---

Given $x \in \mathbb{R}$ and $N \in \mathbb{Z}_{>0}$ let $x_N = \text{sign}(x)\frac{\lfloor 2^N |x| \rfloor}{2^N}$, which corresponds to the operation of keeping the first $N$ bits after zero of a real number $x$.

We now explain informally the steps of the algorithm. In the first step, the algorithm truncates each entry of $Y$ and $X$ by keeping only its first $N$ bits after zero, for some $N \in \mathbb{Z}_{>0}$. This in particular allows to perform finite-precision operations and to call the ELO algorithm in the next step which is designed for integer input. In the second step, the algorithm naturally scales up the truncated data to integer values, that is it scales $Y_N$ by $2^N \hat{Q}$ and $X_N$ by $2^N$. The reason for the additional multiplication of the observation vector $Y$ by $\hat{Q}$ is necessary to make sure the ground truth vector $\beta^*$ can be treated as integer-valued. To see this notice that $Y = X\beta^* + W$ and $Y_N, X_N$ being "close" to $Y, X$ imply

$$2^N \hat{Q} Y_N = 2^N X_N(\hat{Q}\beta^*) + \text{``extra noise terms"} + 2^N \hat{Q}W.$$

Therefore, assuming the control of the magnitude of the extra noise terms, by using the $Q$-rationality assumption and that $\hat{Q}$ is estimated to be a multiple of $Q$, the new ground truth vector becomes $\hat{Q}\beta^*$ which is integer-valued. The final step of the algorithm consist of rescaling now the output of Step 2, to an output which is estimated to be the original $\beta^*$. In the next subsection, we turn this discussion into a provable recovery guarantee.

### Recovery Guarantees for the LBR algorithm

We state now our main result, explicitly stating the assumptions on the parameters, under which the LBR algorithm recovers **exactly** $\beta^*$ from bounded but **adversarial noise** $W$.

**Theorem 2.5.A.** *Under Assumption 1 and assuming $W \in [-\sigma, \sigma]^n$ for some $\sigma \geq 0$, the following holds. Suppose $\hat{Q}$ is a multiple of $Q$, $\hat{R} \geq R$ and*

$$N > \frac{1}{2}(2n + p)\left( 2n + p + 10 \log \hat{Q} + 10 \log \left( 2^N \sigma + \hat{R}p \right) + 20 \log(3(1 + c)np) \right). \quad (3)$$

*For any $\hat{W} \geq \sigma$, the LBR algorithm with input $(Y, X, N, \hat{Q}, \hat{R}, \hat{W})$ terminates with $\hat{\beta}^* = \beta^*$ w.p. $1 - O\left(\frac{1}{np}\right)$ (whp as $p \to +\infty$) and in time polynomial in $n, p, N, \log \hat{R}, \log \hat{W}$ and $\log \hat{Q}$.*

Applying Theorem 2.5.A we establish the following result handling **random noise** $W$.

**Theorem 2.5.B.** *Under Assumption 1 and assuming $W \in \mathbb{R}^n$ is a vector with iid entries generating according to an, independent from $X$, distribution $\mathcal{W}$ on $\mathbb{R}$ with mean zero and variance at most $\sigma^2$ for some $\sigma \geq 0$ the following holds. Suppose that $\hat{Q}$ is a multiple of $Q$, $\hat{R} \geq R$, and*

$$N > \frac{1}{2}(2n + p)\left( 2n + p + 10 \log \hat{Q} + 10 \log \left( 2^N \sqrt{np}\sigma + \hat{R}p \right) + 20 \log(3(1 + c)np) \right). \quad (4)$$

*For any $\hat{W} \geq \sqrt{np}\sigma$ the LBR algorithm with input $(Y, X, N, \hat{Q}, \hat{R}, \hat{W})$ terminates with $\hat{\beta}^* = \beta^*$ w.p. $1 - O\left(\frac{1}{np}\right)$ (whp as $p \to +\infty$) and in time polynomial in $n, p, N, \log \hat{R}, \log \hat{W}$ and $\log \hat{Q}$.*

### Noise tolerance of the LBR algorithm

The assumptions (2) and (4) might make it hard to build an intuition for the truncation level the LBR algorithm provably works. For this reason, in this subsection we *simplify it* and state a Proposition

explicitly mentioning the optimal truncation level and hence characterizing the optimal level of noise that the LBR algorithm can tolerate with $n$ samples.

First note that in the statements of Theorem 2.5.A and Theorem 2.5.B the only parameters that are assumed to grow are $p$ and whichever other parameter is implied to grow because of (2) and (4). Therefore, importantly, $n$ does not necessarily grow to infinity, if for example $N, \frac{1}{\sigma}$ grow appropriately with $p$. That means that Theorem 2.5.A and Theorem 2.5.B imply non-trivial guarantees for *arbitrary sample size* $n$. The proposition below shows that if $\sigma$ is at most exponential in $-(1+\epsilon)\left[\frac{(p+2n)^2}{2n} + (2+\frac{p}{n})\log(RQ)\right]$ for some $\epsilon > 0$, then for appropriately chosen truncation level $N$ the LBR algorithm recovers exactly the vector $\beta^*$ with $n$ samples. In particular, with one sample ($n=1$) LBR algorithm tolerates noise level up to exponential in $-(1+\epsilon)\left[p^2/2 + (2+p)\log(QR)\right]$ for some $\epsilon > 0$. On the other hand, if $n = \Theta(p)$ and $\log(RQ) = o(p)$, the LBR algorithm tolerates noise level up to exponential in $-O(p)$.

**Proposition 2.6.** *Under Assumption 1 and assuming $W \in \mathbb{R}^n$ is a vector with iid entries generating according to an, independent from $X$, distribution $\mathcal{W}$ on $\mathbb{R}$ with mean zero and variance at most $\sigma^2$ for some $\sigma \geq 0$, the following holds. Suppose for some $\epsilon > 0$, $\sigma \leq 2^{-(1+\epsilon)\left[\frac{(p+2n)^2}{2n} + (2+\frac{p}{n})\log(RQ)\right]}$. Then the LBR algorithm with input $Y, X, \hat{Q} = Q, \hat{R} = R, \hat{W}_\infty = 1$, features $p \geq \frac{300}{\epsilon}\log\left(\frac{300}{(1+c)\epsilon}\right)$ and $N$ satisfying $\log\left(\frac{1}{\sigma}\right) \geq N \geq (1+\epsilon)\left[\frac{(p+2n)^2}{2n} + (2+\frac{p}{n})\log(RQ)\right]$, terminates with $\hat{\beta}^* = \beta^*$ w.p. $1 - O\left(\frac{1}{np}\right)$ (whp as $p \to +\infty$) and in time polynomial in $n, p, N, \log\hat{R}, \log\hat{W}$ and $\log\hat{Q}$.*

### Information Theoretic Bounds

In this subsection, we discuss the maximum noise that can be tolerated information-theoretically in recovering a $\beta^* \in [-R, R]^p$ satisfying the $Q$-rationality assumption. We establish that under Gaussian white noise, any successful recovery mechanism can tolerate noise level at most exponentially small in $-[p\log(QR)/n]$.

**Proposition 2.7.** *Suppose that $X \in \mathbb{R}^{n \times p}$ is a vector with iid entries following a continuous distribution $\mathcal{D}$ with $\mathbb{E}[|V|] < +\infty$, where $V \stackrel{d}{=} \mathcal{D}$, $\beta^* \in [-R, R]^p$ satisfies the $Q$-rationality assumption, $W \in \mathbb{R}^n$ has iid $N(0, \sigma^2)$ entries and $Y = X\beta^* + W$. Suppose furthermore that $\sigma > R(np)^3 \left(2^{\frac{2p\log(2QR+1)}{n}} - 1\right)^{-\frac{1}{2}}$. Then there is **no** mechanism which, whp as $p \to +\infty$, recovers **exactly** $\beta^*$ with knowledge of $Y, X, Q, R, \sigma$. That is, for any $\hat{\beta}^* = \hat{\beta}^*(Y, X, Q, R, \sigma)$ we have*

$$\limsup_{p \to +\infty} \mathbb{P}\left(\hat{\beta}^* = \beta^*\right) < 1.$$

### Sharp Optimality of the LBR Algorithm

Using Propositions 2.6 and 2.7 the following **sharp** result is established.

**Proposition 2.8.** *Under Assumptions 1 where $W \in \mathbb{R}^n$ is a vector with iid $N(0, \sigma^2)$ entries the following holds. Suppose that $n = o\left(\frac{p}{\log p}\right)$ and $RQ = 2^{\omega(p)}$. Then for $\sigma_0 := 2^{-\frac{p\log(RQ)}{n}}$ and $\epsilon > 0$:*

- *if $\sigma > \sigma_0^{1-\epsilon}$, then the w.h.p. exact recovery of $\beta^*$ from the knowledge of $Y, X, Q, R, \sigma$ is impossible.*

- *if $\sigma < \sigma_0^{1+\epsilon}$, then the w.h.p. exact recovery of $\beta^*$ from the knowledge of $Y, X, Q, R, \sigma$ is possible by the LBR algorithm.*

## 3 Synthetic Experiments

In this section we present an experimental analysis of the ELO and LBR algorithms.

*ELO algorithm:* We focus on $p = 30$ features sample sizes $n = 1, n = 10$ and $n = 30$, $R = 100$ and zero-noise $W = 0$. Each entry of $\beta^*$ is iid $\text{Unif}(\{1, 2, \ldots, R = 100\})$. For 10 values of $\alpha \in (0, 3)$,

specifically $\alpha \in \{0.25, 0.5, 0.75, 1, 1.3, 1.6, 1.9, 2.25, 2.5, 2.75\}$, we generate the entries of $X$ iid $\mathrm{Unif}\left(\{1, 2, 3, \dots, 2^N\}\right)$ for $N = \frac{p^2}{2\alpha n}$. For each combination of $n, \alpha$ we generate 20 independent instances of inputs. We plot in Figure 1 the fractions of instances where the output of the ELO algorithm outputs exactly $\beta^*$ and the average termination time of the algorithm.

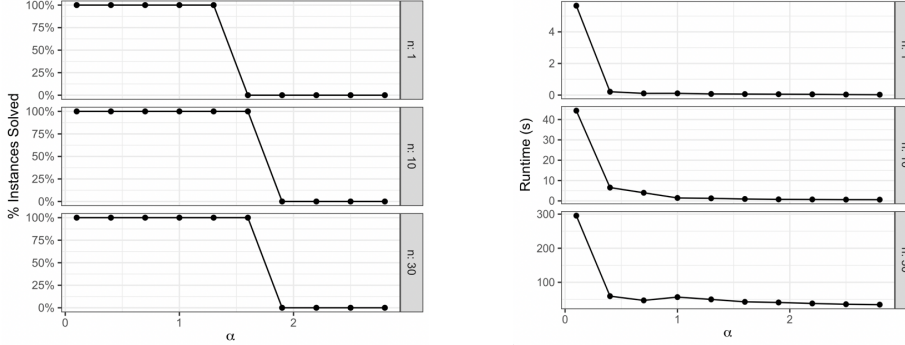

Figure 1: Average performance and runtime of ELO over 20 instances with $p = 30$ features and $n = 1, 10, 30$ samples.

**Comments:** First, we observe that importantly the algorithm recovers the vectors correctly on all $\alpha < 1$-instances with $p = 30$ features, even if our theoretical guarantees are only for large enough $p$. Second, Theorem 2.1 implies that if $N > (2n + p)^2 / 2n$ and large $p$, ELO recovers $\beta^*$, with high probability. In the experiments we observe that indeed ELO algorithm works in that regime, as then $\alpha = \frac{p^2}{2nN} < 1$. Also the experiments show that ELO works for larger values of $\alpha$. Finally, the termination time of the algorithm was on average 1 minute and worst case 5 minutes, granting it reasonable for many applications.

*LBR algorithm:* We focus on $p = 30$ features, $n = 10$ samples, $Q = 1$ and $R = 100$. We generate each entry of $\beta^*$ w.p. 0.5 equal to zero and w.p. 0.5, $\mathrm{Unif}\left(\{1, 2, \dots, R = 100\}\right)$. We generate the entries of $X$ iid $U(0, 1)$ and of $W$ iid $U(-\sigma, \sigma)$ for $\sigma \in \{0, e^{-20}, e^{-12}, e^{-4}\}$. We generate 20 independent instances for any combination of $\sigma$ and truncation level $N$. We plot the fraction of instances where the output of LBR algorithm is exactly $\beta^*$.

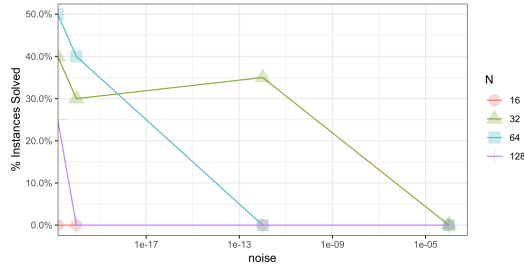

Figure 2: Average performance of LBR algorithm for various noise and truncation levels.

**Comments:** The experiments show that, first LBR works correctly in many cases for the moderate value of $p = 30$ and second that there is indeed an appropriate tuned truncation level $(2n + p)^2 / 2n < N < \log(1/\sigma)$ for which LBR succeeds. The latter is in exact agreement with Proposition 2.6.

### Acknowledgments

The authors would like to gratefully aknowledge the work of Patricio Foncea and Andrew Zheng on performing the synthetic experiments for the ELO and LBR algorithms, as part of a project for a graduate-level class at MIT, during Spring 2018.

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
