[Reviews · NeurIPS 2018]

Reviewer 1



* Summary This paper studies a high-dimensional regression problem through applying the Lattice-based technique The technique restricts the true parameters \beta^* as rational numbers and represent them as division of integers, and obtain the true parameters as an output of the Lagarias-Odlyzko algorithm. Their approach enables us to obtain the true parameter with ultra high-dimensional case in which the previous approach could not prove theoretical guarantees. * Comment The idea of this paper is quite interesting for me. Though the Q-rationality assumption is restrictive a little, it seems an appropriate way to simplify the regression problem and obtain a sharp result. However, I have mainly two comments for some faults of this paper. 1. Assumption on noise W is quite restrictive. The authors assume that the noise level decreases rapidly as n increases, and this is a severe condition for high-dimensional regression. (For an instance, this paper assume \sigma = o(n^{-1}\hat{Q}^{-1}) in Theorem 2.5.A.) In the field of high-dimensional statistics, it is always assumed that \sigma=O(1), so the setting of this paper is not acceptable for the statistics. 2. Numerical evidence This paper lacks numerical evidences to support usefulness of the result of this paper. Since this paper provides theoretical results by applying a new framework which is not common in the fields of high-dimensional statistics, the authors should show validity and usefulness of the new approach by numerics. Also, I have some minor comments. -The limit condition (eq.(2)) is difficult to interpret. What N works for? -Typo: in line 253, \beta^*_i should be \beta_i ?

Reviewer 2



The paper presents a novel method of exactly recovering a vector of coefficients in high-dimensional linear regression, with high probability as the dimension goes to infinity. The method assumes that the correct coefficients come from a finite discrete set of bounded rational values, but it does not - as is commonplace - assume that the coefficient vector is sparse. To achieve this, the authors extend a classical algorithm for lattice basis reduction. Crucially, this approach does not require the sample size to grow with the dimension, thus in certain cases the algorithm is able to recover the exact coefficient vector from just a single sample (with the dimension sufficiently large). A novel connection between high-dimensional linear regression and lattice basis reduction is the main strength of the paper. The ideas and algorithms in the paper are presented clearly. The main limitation of the algorithm is the fact that all of the guarantees hold „with high probability as dimension p goes to infinity”. No finite dimension guarantees are offered, which raises the question of whether the proposed algorithm only achieves the desired behavior for combinatorially large dimensions. That would make the „polynomial runtime” guarantee somewhat misleading (since the runtime depends on p). It is worth noting that despite the authors’ claim that taking dimension p to infinity is „very popular in the literature”, after looking at some of the cited works I noticed that typically at least some finite dimension guarantees are offered in these works. Thus, to address this, in addtion to „w.h.p. when p tends to infinity”, the authors could - if at all possible - state the guarantees as „w.p. at least 1-delta for p larger than…”, for example. Minor comments: - line 76: „so that to” -> „so as to” - line 212: „can be applied,.” -> „can be applied.”

Reviewer 3



This paper introduces the problem of high-dimensional regression with a structural assumption on the parameters. Formally, X and y are observed such that either y=Xb* or y=Xb*+noise. This problem is ill-posed without further assumptions on b* in high dimension. Here, the assumption is that b* has rational coefficients, with a fixed bounded denominator. A new algorithm is proposed, related to the LLL basis reduction algorithm, and analysed, showing that some amount of noise can be tolerated in this problem. I think that the result is interesting, and as far as I can tell, correct. Maybe the main weakness of the paper is that it is not very clear how useful and well-motivated this problem is. However, the technical contribution is important.